# ONLINE LEARNING FOR SUPERVISED DIMENSION REDUCTION

## ABSTRACT

Online learning has attracted great attention due to the increasing demand for systems that have the ability of learning and evolving. When the data to be processed is also high dimensional and dimension reduction is necessary for visualization or prediction enhancement, online dimension reduction will play an essential role. The purpose of this paper is to propose new online learning approaches for supervised dimension reduction. Our first algorithm is motivated by adapting the sliced inverse regression (SIR), a pioneer and effective algorithm for supervised dimension reduction, and making it implementable in an incremental manner. The new algorithm, called incremental sliced inverse regression (ISIR), is able to update the subspace of significant factors with intrinsic lower dimensionality fast and efficiently when new observations come in. We also refine the algorithm by using an overlapping technique and develop an incremental overlapping sliced inverse regression (IOSIR) algorithm. We verify the effectiveness and efficiency of both algorithms by simulations and real data applications.

## 1 INTRODUCTION

Dimension reduction aims to explore low dimensional representation for high dimensional data. It helps to promote our understanding of the data structure through visualization and enhance the predictive performance of machine learning algorithms by preventing the "curse of dimensionality". Therefore, as high dimensional data become ubiquitous in modern sciences, dimension reduction methods are playing more and more important roles in data analysis. Dimension reduction algorithms can be either unsupervised or supervised. Principle component analysis (PCA) might be the most popular unsupervised dimension reduction method. Other unsupervised dimension reduction methods include the kernel PCA, multidimensional scaling, and manifold learning based methods such as isometric mapping and local linear embedding. Unlike unsupervised dimension reduction, supervised dimension reduction involves a response variable. It finds the intrinsic lower-dimensional representations that are relevant to the prediction of the response values. Supervised dimension reduction methods can date back to the well known linear discriminant analysis (LDA) while its blossom occurred in the last twenty years. Many approaches have been proposed and successfully applied in various scientific domains; see Li (1991); Cook & Weisberg (1991); Li (1992); Xia et al. (2002); Setodji & Cook (2004); Fukumizu et al. (2004); Li & Yin (2008); Wu (2008); Wu et al. (2009); Cook & Zhang (2014) and the references therein.

We are in a big data era and facing the challenges of big data processing, thanks to the fast development of modern information technology. Among others, two primary challenges are the big volume and fast velocity of the data. When a data set is too big to be stored in a single machine or when the data arrives in real time and information update is needed frequently, analysis of the data in an online manner is necessary and efficient. If the data is simultaneously big and high dimensional, it becomes necessary to develop online learning approaches for dimension reduction. As PCA and LDA are the most wildly used dimension reduction techniques, a bunch of PCA-based and LDA-based online dimension reduction algorithms has been proposed. Incremental PCA have been described in Hall et al. (1998; 2000); Weng et al. (2003); Zhao et al. (2006); Rodriguez & Wohlberg (2014). Incremental LDA have been developed in Pang et al. (2005); Zhao & Yuen (2008); Kim et al. (2007); Chu et al. (2015). Other strategies like QR decomposition or SVD have also been used in Chandrasekaran et al. (1997); Ye et al. (2005); Ren & Dai (2010).

In this paper, our purpose is to propose a new online learning approach for supervised dimension reduction. Our motivation is to implement the sliced inverse regression (SIR) in an incremental manner. SIR was proposed in Li (1991) and has become one of the most efficient supervised dimension reduction method. SIR and its refined versions have been found successful in many scientific areas such as bioinformatics, hyperspectral image analysis, and physics; see Cook (1994); Belhumeur et al. (1997); Becker & Fried (2003); Elnitski et al. (2003); Gannoun et al. (2004); He et al. (2003); Antoniadis et al. (2003); Li & Li (2004); Dai et al. (2006); Zhang et al. (2016). SIR can be implemented by solving an generalized eigen-decomposition problem, $\mathbf{\Gamma}\boldsymbol{\beta} = \lambda\mathbf{\Sigma}\boldsymbol{\beta}$, where $\mathbf{\Gamma}$ is a matrix depending on the response variable (whose definition is described in the next section) and $\mathbf{\Sigma}$ is the covariance matrix. To make it implementable in an online manner we rewrite it as standard eigen-decomposition problem $\mathbf{\Sigma}^{-\frac{1}{2}}\mathbf{\Gamma}\mathbf{\Sigma}^{-\frac{1}{2}}\boldsymbol{\eta} = \lambda\boldsymbol{\eta}$ where $\boldsymbol{\eta} = \mathbf{\Sigma}^{\frac{1}{2}}\boldsymbol{\beta}$ and adopt the ideas from incremental PCA. We need to overcome two main challenges in this process. First, how do we transform the data so that they are appropriate for the transformed PCA problem? Note that simply normalizing the data does not work. Second, online update of $\mathbf{\Sigma}^{-\frac{1}{2}}$, if not impossible, seems very difficult. The first contribution of this paper is to overcome these difficulties and design a workable incremental SIR method. Our second contribution will be to refine the method by an overlapping technique and design an incremental overlapping SIR algorithm.

The rest of this paper is arranged as follows. We review SIR algorithm in Section 2 and the incremental PCA algorithm in Section 3. We propose the incremental SIR algorithm in Section 4 and refine it in Section 5. Simulations are done in Section 6. We close with discussions in Section 7.

## 2 SLICED INVERSE REGRESSION

The goal of supervised dimension reduction is to find an intrinsic lower-dimensional subspace that contains all the information to predict the response variable. Assume a multivariate predictor $\mathbf{x} = (x_1, x_2, \ldots, x_p)^\top \in \mathbb{R}^p$ and a scalar response $y$ are linked by a semi-parametric regression model

$$y = f(\boldsymbol{\beta}_1^\top\mathbf{x}, \boldsymbol{\beta}_2^\top\mathbf{x}, \ldots, \boldsymbol{\beta}_K^\top\mathbf{x}, \epsilon), \tag{1}$$

where $\boldsymbol{\beta}_k \in \mathbb{R}^p$ is a $p \times 1$ vector and $\epsilon$ is the error term independent of $\mathbf{x}$. It implies

$$y \perp\!\!\!\perp \mathbf{x} | \mathbf{B}^\top\mathbf{x} \tag{2}$$

where $\perp\!\!\!\perp$ denotes "statistical independence" and $\mathbf{B} = (\boldsymbol{\beta}_1, \boldsymbol{\beta}_2, \ldots, \boldsymbol{\beta}_K)$ is a $p \times K$ matrix. The column space of $\mathbf{B}$ is called the effective dimension reduction (EDR) space and each $\boldsymbol{\beta}_i$ is an EDR direction. Note $\mathbf{B}^\top\mathbf{x}$ contains all information for the prediction of $y$. The purpose of supervised dimension reduction is to learn the EDR directions from data.

Unlike the classical regression problem which regresses $y$ against $\mathbf{x}$, sliced inverse regression considers regressing $\mathbf{x}$ against $y$. With the semi-parametric model (1) and the assumption that $\mathbf{x}$ follows an elliptical contour distribution (e.g., normal distribution), it was proved in Li (1991) that the centered regression curve $\mathbf{E}[\mathbf{x}|y] - \mathbf{E}[\mathbf{x}]$ falls into the $K$-dimensional subspace spanned by EDR directions $\mathbf{\Sigma}\boldsymbol{\beta}_k, k = 1, \ldots, K$. Consequently, all or part of the EDR directions can be recovered by solving a generalized eigenvalue decomposition problem:

$$\mathbf{\Gamma}\boldsymbol{\beta} = \lambda\mathbf{\Sigma}\boldsymbol{\beta} \tag{3}$$

where

$$\mathbf{\Gamma} = \mathbf{E}\left[\left(\mathbf{E}[\mathbf{x}|y] - \mathbf{E}[\mathbf{x}]\right)\left(\mathbf{E}[\mathbf{x}|y] - \mathbf{E}[\mathbf{x}]\right)^\top\right]$$

is the covariance matrix of inverse regression curve $\mathbf{E}[\mathbf{x}|y]$, $\mathbf{\Sigma}$ is the covariance matrix of $\mathbf{x}$. Each eigenvector associated with a non-zero eigenvalue is an EDR direction.

Given i.i.d observations $\{(\mathbf{x}_i, y_i)\}_{i=1}^n$, SIR algorithm can be implemented as follows:

1) Compute sample mean $\bar{\mathbf{x}} = \frac{1}{n}\sum_{i=1}^n \mathbf{x}_i$ and the sample covariance matrix

$$\widehat{\mathbf{\Sigma}} = \frac{1}{n}\sum_{i=1}^n (\mathbf{x}_i - \bar{\mathbf{x}})(\mathbf{x}_i - \bar{\mathbf{x}})^\top.$$

2) Bin the observations into $H$ slices according to $y$ values. For each slice $s_h$, $h = 1, \ldots, H$, compute the sample probability $\widehat{p}_h = \frac{n_h}{n}$ and the sample slice mean $\widehat{\mathbf{m}}_h = \frac{1}{n_h} \sum_{y_i \in s_h} \mathbf{x}_i$. The matrix $\boldsymbol{\Gamma}$ is estimated by

$$\widehat{\boldsymbol{\Gamma}} = \sum_{h=1}^{H} \widehat{p}_h \left( \widehat{\mathbf{m}}_h - \bar{\mathbf{x}} \right) \left( \widehat{\mathbf{m}}_h - \bar{\mathbf{x}} \right)^{\top}.$$

3) Solve the generalized eigen-decomposition problem

$$\widehat{\boldsymbol{\Gamma}}\widehat{B} = \widehat{\boldsymbol{\Sigma}}\widehat{B}\boldsymbol{\Lambda}.$$

The EDR directions are estimated by the top $K$ eigenvectors $\widehat{\boldsymbol{\beta}}_k, k = 1, 2, \ldots, K$.

This algorithm is not very sensitive to the choice of parameter $H$ provided it is sufficiently larger than $K$ while not greater than $\frac{n}{2}$. Root-$n$ consistency is usually promised. It is suggested samples are evenly distributed into the $H$ slices for the best performance.

# 3 INCREMENTAL PCA

PCA looks for directions along which the data have the largest variances. It is implemented by solving an eigen-decomposition problem

$$\widehat{\boldsymbol{\Sigma}}\mathbf{u} = \lambda\mathbf{u}. \tag{4}$$

The principal components are the eigenvectors corresponding to largest eigenvalues. Throughout this paper, we assume all eigenvalues are arranged in a descending order, i.e., $\lambda_1 \geq \lambda_2 \geq \ldots \geq \lambda_p$. Suppose that we need to retain the top $K$ principal components. Denote $\mathbf{U}_K = [\mathbf{u}_1, \ldots, \mathbf{u}_K]$, $\boldsymbol{\Lambda}_K = \mathrm{diag}(\lambda_1, \ldots, \lambda_K)$. We have a reduced system $\widehat{\boldsymbol{\Sigma}}\mathbf{U}_K = \mathbf{U}_K\boldsymbol{\Lambda}_K$.

In incremental PCA, after receiving a new coming observation $\mathbf{x_0}$, we need to update the reduced eigen-system to a new one

$$\widehat{\boldsymbol{\Sigma}}'\mathbf{U}'_K \approx \mathbf{U}'_K\boldsymbol{\Lambda}'_K. \tag{5}$$

A "=" is generally impossible unless $\lambda_{K+1} = \ldots = \lambda_p = 0$ in (4).

The idea of updating the system in Hall et al. (1998) is as follows. Compute a residual vector

$$\mathbf{v} = (\mathbf{x}_0 - \bar{\mathbf{x}}') - \mathbf{U}_K\mathbf{U}_K^{\top}(\mathbf{x}_0 - \bar{\mathbf{x}}')$$

where $\bar{\mathbf{x}}'$ is the mean of all observations (including $\mathbf{x}_0$). It defines the component of $\mathbf{x}_0$ that is perpendicular with the subspace defined by $\mathbf{U}_K$. If $\mathbf{x_0}$ lies exactly within the current eigenspace, then the residual vector is zero and there is no need to update the system. Otherwise, we normalize $\mathbf{v}$ to obtain $\bar{\mathbf{v}} = \frac{\mathbf{v}}{\|\mathbf{v}\|}$. We may reasonably assume each column vector of $\mathbf{U}'_K$ is a linear combination of column vectors of $\mathbf{U}_K$ and $\bar{\mathbf{v}}$. (Note this is exactly true if $\lambda_{K+1} = \ldots = \lambda_p = 0$.) This allows us to write

$$[\mathbf{U}'_K, \mathbf{u}'_{K+1}] = [\mathbf{U}_K, \bar{\mathbf{v}}]\mathbf{R}.$$

where $\mathbf{R}$ is a $(K+1) \times (K+1)$ rotation matrix and $\mathbf{u}'_{K+1}$ is an approximation of the $(K+1)$th eigenvector of $\widehat{\boldsymbol{\Sigma}}'$. So we have

$$\widehat{\boldsymbol{\Sigma}}'[\mathbf{U}_K, \bar{\mathbf{v}}]\mathbf{R} = [\mathbf{U}_K, \bar{\mathbf{v}}]\mathbf{R}\boldsymbol{\Lambda}'_{K+1},$$

which is equivalent to

$$[\mathbf{U}_K, \bar{\mathbf{v}}]^{\top}\widehat{\boldsymbol{\Sigma}}'[\mathbf{U}_K, \widehat{\mathbf{v}}]\mathbf{R} = \mathbf{R}\boldsymbol{\Lambda}'_{K+1}.$$

This is an eigen-decomposition problem of dimensionality $K + 1 \ll p$. It solves the rotation matrix $\mathbf{R}$ and allows us to update principal components to $\mathbf{U}'_K$, given by the first $K$ column of $[\mathbf{U}_K, \bar{\mathbf{v}}]\mathbf{R}$. If we need to increase the number of principal components, we can just update the system to $K' = K + 1$ and $\mathbf{U}'_{K'} = [\mathbf{U}'_K, \mathbf{u}'_{K+1}]$. This incremental PCA algorithm was shown convergent to a stable solution when the sample size increases (Hall et al., 1998).

## 4 INCREMENTAL SIR

Our idea to develop the incremental sliced inverse regression (ISIR) is motivated by reformulating SIR problem to a PCA problem. To this end, we define $\boldsymbol{\eta} = \boldsymbol{\Sigma}^{\frac{1}{2}}\boldsymbol{\beta}$, called the standardized EDR direction, and rewrite the generalized eigen-decomposition problem (3) as an eigen-decomposition problem

$$\boldsymbol{\Sigma}^{-\frac{1}{2}}\boldsymbol{\Gamma}\boldsymbol{\Sigma}^{-\frac{1}{2}}\boldsymbol{\eta} = \lambda\boldsymbol{\eta}. \tag{6}$$

Note that $\boldsymbol{\Sigma}^{-\frac{1}{2}}\boldsymbol{\Gamma}\boldsymbol{\Sigma}^{-\frac{1}{2}}$ is the covariance matrix of $\boldsymbol{\Sigma}^{-\frac{1}{2}}\mathbf{E}[\mathbf{x}|y]$. So (6) can be regarded as a PCA problem with data collected for $\boldsymbol{\Sigma}^{-\frac{1}{2}}\mathbf{E}[\mathbf{x}|y]$. To apply the ideas from IPCA to this transformed PCA problem, however, is not as direct as it looks like. We face two main challenges. First, when a new observation $(\mathbf{x}_0, y_0)$ is received, we need to transform it to an observation for the standardized inverse regression curve. This is different from simply standardizing the data. Second, conceptually, we need to update $\boldsymbol{\Sigma}^{-\frac{1}{2}}$ in an online manner in order to standardize the data. This does not seem feasible. In the following, we will describe in detail how we address these challenges and make the ISIR implementable.

Suppose we have $n$ observations in hand with well defined sample slice probabilities $\widehat{p}_h$ and means $(\widehat{\mathbf{m}}_h, \bar{y}_h)$ for $h = 1, \ldots, H$, and the eigenvectors $\widehat{\mathbf{B}} = [\widehat{\boldsymbol{\beta}}_1, \ldots, \widehat{\boldsymbol{\beta}}_K]$ of the generalized eigen-decomposition problem $\widehat{\boldsymbol{\Gamma}} = \lambda\widehat{\boldsymbol{\Sigma}}\boldsymbol{\beta}$. Then with $\boldsymbol{\Lambda}_K = \mathrm{diag}(\lambda_1, \ldots, \lambda_K)$, we have

$$\widehat{\boldsymbol{\Gamma}}\widehat{\mathbf{B}} = \widehat{\boldsymbol{\Sigma}}\widehat{\mathbf{B}}\boldsymbol{\Lambda}_K.$$

Denote $\boldsymbol{\Xi} = \widehat{\boldsymbol{\Sigma}}^{\frac{1}{2}}\widehat{\mathbf{B}}$. We have

$$\widehat{\boldsymbol{\Sigma}}^{-\frac{1}{2}}\widehat{\boldsymbol{\Gamma}}\widehat{\boldsymbol{\Sigma}}^{-\frac{1}{2}}\boldsymbol{\Xi} = \boldsymbol{\Xi}\boldsymbol{\Lambda}_K. \tag{7}$$

When we have a new observation $(\mathbf{x}_0, y_0)$, we first locate which slice it belongs to according to the distances from $y_0$ to sample slice mean values $\bar{y}_h$ of the response variable. Let us suppose the distance from $y_0$ to $\bar{y}_k$ is the smallest. So we place the new observation into the slice $k$ and update sample slice probabilities by $\widehat{p}'_h = \frac{n\widehat{p}_h}{n+1}$ for $h \neq k$ and $p'_k = \frac{np'_k+1}{n+1}$. Let $n_k = np_k$ be the number of observations in slice $k$ before receiving the new observation. For slice mean values we update

$$\widehat{\mathbf{m}}'_k = \frac{n_k}{n_k+1}\widehat{\mathbf{m}}_k + \frac{1}{n_k+1}\mathbf{x}_0$$

for slice $k$ only. We can regard $\mathbf{z}_0 = \widehat{\boldsymbol{\Sigma}}^{-\frac{1}{2}}\mathbf{m}'_k$ as a new observation for the standardized inverse regression curve $\boldsymbol{\Sigma}^{-\frac{1}{2}}\mathbf{E}[\mathbf{x}|y]$. Following the idea of IPCA, we define a residual vector

$$\mathbf{v} = \left(\mathbf{z}_0 - \widehat{\boldsymbol{\Sigma}}^{-\frac{1}{2}}\bar{\mathbf{x}}'\right) - \boldsymbol{\Xi}\boldsymbol{\Xi}^{\top}\left(\mathbf{z}_0 - \widehat{\boldsymbol{\Sigma}}^{-\frac{1}{2}}\bar{\mathbf{x}}'\right)$$

and normalize it to $\bar{\mathbf{v}} = \frac{\mathbf{v}}{\|\mathbf{v}\|}$ when $\mathbf{v}$ is not zero. To update the eigen-decomposition system to

$$\widehat{\boldsymbol{\Sigma}'}^{-\frac{1}{2}}\widehat{\boldsymbol{\Gamma}'}\widehat{\boldsymbol{\Sigma}'}^{-\frac{1}{2}}\boldsymbol{\Xi}' = \boldsymbol{\Xi}'\boldsymbol{\Lambda}'_K, \tag{8}$$

we assume $[\boldsymbol{\Xi}', \boldsymbol{\eta}'_{K+1}] = [\boldsymbol{\Xi}, \bar{\mathbf{v}}]\mathbf{R}$ with $\mathbf{R}$ being a $(K+1) \times (K+1)$ rotation matrix and $\boldsymbol{\eta}'_{K+1}$ the $(K+1)$th eigenvector of $\widehat{\boldsymbol{\Sigma}'}^{-\frac{1}{2}}\widehat{\boldsymbol{\Gamma}'}\widehat{\boldsymbol{\Sigma}'}^{-\frac{1}{2}}$. So we have

$$\widehat{\boldsymbol{\Sigma}'}^{-\frac{1}{2}}\widehat{\boldsymbol{\Gamma}'}\widehat{\boldsymbol{\Sigma}'}^{-\frac{1}{2}}[\boldsymbol{\Xi}, \bar{\mathbf{v}}]\mathbf{R} = [\boldsymbol{\Xi}, \bar{\mathbf{v}}]\mathbf{R}\boldsymbol{\Lambda}'_{K+1}$$

where $\boldsymbol{\Lambda}'_{K+1} = \mathrm{diag}(\boldsymbol{\Lambda}'_K, \lambda'_{K+1})$ and $\lambda'_{K+1}$ is the $(K+1)$th eigenvalue. Multiplying both sides by $[\boldsymbol{\Xi}, \bar{\mathbf{v}}]^{\top}$, we obtain

$$\left(\widehat{\boldsymbol{\Sigma}'}^{-\frac{1}{2}}[\boldsymbol{\Xi}, \bar{\mathbf{v}}]\right)^{\top}\widehat{\boldsymbol{\Gamma}'}\left(\widehat{\boldsymbol{\Sigma}'}^{-\frac{1}{2}}[\boldsymbol{\Xi}, \bar{\mathbf{v}}]\right)\mathbf{R} = \mathbf{R}\boldsymbol{\Lambda}'_{K+1}. \tag{9}$$

Note, however, since $\widehat{\boldsymbol{\Sigma}'}^{-\frac{1}{2}}$ cannot be easily updated, we have to avoid using it. To overcome this challenge, we notice that

$$\widehat{\boldsymbol{\Sigma}'} = \frac{n}{n+1}\widehat{\boldsymbol{\Sigma}'} + \frac{n}{(n+1)^2}(\mathbf{x_0} - \bar{\mathbf{x}})(\mathbf{x_0} - \bar{\mathbf{x}})^{\top} \tag{10}$$

and the well known Sherman-Morisson formula allows us to update the inverse covariance matrix

$$\widehat{\boldsymbol{\Sigma}'}^{-1} = \frac{n+1}{n}\widehat{\boldsymbol{\Sigma}}^{-1} - \frac{\frac{1}{n}\widehat{\boldsymbol{\Sigma}}^{-1}(\mathbf{x}_0 - \bar{\mathbf{x}})(\mathbf{x}_0 - \bar{\mathbf{x}})^\top\widehat{\boldsymbol{\Sigma}}^{-1}}{1 + \frac{1}{n+1}(\mathbf{x}_0 - \bar{\mathbf{x}})^\top\widehat{\boldsymbol{\Sigma}}^{-1}(\mathbf{x}_0 - \bar{\mathbf{x}})}. \tag{11}$$

If we store $\widehat{\boldsymbol{\Sigma}}^{-1}$ and update it incrementally, we can approximate the quantities in (9) as follows:

$$\widehat{\boldsymbol{\Sigma}'}^{-\frac{1}{2}}\boldsymbol{\Xi} \approx \widehat{\boldsymbol{\Sigma}}^{-\frac{1}{2}}\boldsymbol{\Xi} = \widehat{\mathbf{B}},$$

$$\widehat{\boldsymbol{\Sigma}'}^{-\frac{1}{2}}\mathbf{v} \approx \widehat{\boldsymbol{\Sigma}}^{-1}(\widehat{\mathbf{m}}'_k - \bar{\mathbf{x}}') - \widehat{\mathbf{B}}\widehat{\mathbf{B}}^\top(\widehat{\mathbf{m}}'_k - \bar{\mathbf{x}}'),$$

$$\|\mathbf{v}\|^2 = (\widehat{\mathbf{m}}'_k - \bar{\mathbf{x}}')^\top\left(\widehat{\boldsymbol{\Sigma}}^{-1} - \widehat{\mathbf{B}}\widehat{\mathbf{B}}^\top\right)(\widehat{\mathbf{m}}'_k - \bar{\mathbf{x}}'),$$

$$\widetilde{\mathbf{v}} = \widehat{\boldsymbol{\Sigma}'}^{-\frac{1}{2}}\bar{\mathbf{v}} = \frac{\widehat{\boldsymbol{\Sigma}'}^{-\frac{1}{2}}\mathbf{v}}{\|\mathbf{v}\|}$$

$$\approx \frac{\left(\widehat{\boldsymbol{\Sigma}}^{-1} - \widehat{\mathbf{B}}\widehat{\mathbf{B}}^\top\right)(\widehat{\mathbf{m}}'_k - \bar{\mathbf{x}}')}{\sqrt{(\widehat{\mathbf{m}}'_k - \bar{\mathbf{x}}')^\top\left(\widehat{\boldsymbol{\Sigma}}^{-1} - \widehat{\mathbf{B}}\widehat{\mathbf{B}}^\top\right)(\widehat{\mathbf{m}}'_k - \bar{\mathbf{x}}')}}.$$

So the problem (9) is approximated by

$$\left[\widehat{\mathbf{B}}, \widetilde{\mathbf{v}}\right]^\top \widehat{\boldsymbol{\Gamma}'}\left[\widehat{\mathbf{B}}, \widetilde{\mathbf{v}}\right]\mathbf{R} = \mathbf{R}\boldsymbol{\Lambda}'_{K+1}. \tag{12}$$

Finally notice that the new EDR space $\widehat{\mathbf{B}}' = \widehat{\boldsymbol{\Sigma}'}^{-\frac{1}{2}}\boldsymbol{\Xi}'$ is the first $K$ columns of $\widehat{\boldsymbol{\Sigma}'}^{-\frac{1}{2}}[\boldsymbol{\Xi}', \boldsymbol{\eta}'_{K+1}] = \widehat{\boldsymbol{\Sigma}'}^{-\frac{1}{2}}[\boldsymbol{\Xi}, \bar{\mathbf{v}}]\mathbf{R}$ and can be approximated by the first $K$ columns of $[\widehat{\mathbf{B}}, \widetilde{\mathbf{v}}]\mathbf{R}$.

Note that we avoided updating the inverse square root of the covariance matrix by using the approximation $\widehat{\boldsymbol{\Sigma}}^{-\frac{1}{2}} \approx \widehat{\boldsymbol{\Sigma}'}^{-\frac{1}{2}}$. This approximation can be very accurate when $n$ is large enough because both converge to $\boldsymbol{\Sigma}^{-\frac{1}{2}}$. Therefore, we may expect the convergence of ISIR as a corollary of the convergence of IPCA. However, when $n$ is small, the approximation may be less accurate and result in larger difference between EDR spaces estimated by ISIR and SIR. So we recommend that ISIR be used with a warm start, that is, using SIR first on a small amount of data before using ISIR.

In terms of memory, the primary requirement is the storage of $\widehat{\boldsymbol{\Sigma}}^{-1}$, the slice mean matrix $\widehat{\mathbf{M}} = [\widehat{\mathbf{m}}_1, \ldots, \widehat{\mathbf{m}}_H]$, and the EDR space $\widehat{\mathbf{B}}$. So the memory requirement is $O(p^2 + pH + pK)$. As for the computational complexity, notice that the update of $\widehat{\boldsymbol{\Sigma}'}^{-1}$ in (11) requires the calculation of $\widehat{\boldsymbol{\Sigma}}^{-1}(\mathbf{x}_0 - \bar{\mathbf{x}})$ and matrix addition and thus has a complexity of $O(p^2)$. Since we need to store $\widehat{\mathbf{M}}$ and update it sequentially, it is not efficient to store and update $\widehat{\boldsymbol{\Gamma}'}$ for either memory or computation consideration. Instead, we use the fact $\widehat{\boldsymbol{\Gamma}'} = \widehat{\mathbf{M}'}\widehat{\mathbf{P}'}\widehat{\mathbf{M}'}^\top$ where $\widehat{\mathbf{P}'} = \mathrm{diag}(\widehat{p}'_1, \ldots, \widehat{p}'_H)$ and write

$$[\widehat{\mathbf{B}}, \widetilde{\mathbf{v}}]^\top\widehat{\boldsymbol{\Gamma}'}[\widehat{\mathbf{B}}, \widetilde{\mathbf{v}}] = \left[\widehat{\mathbf{B}}^\top\widehat{\mathbf{M}'}, \widetilde{\mathbf{v}}^\top\widehat{\mathbf{M}'}\right]\widehat{\mathbf{P}'}\left[\widehat{\mathbf{B}}^\top\widehat{\mathbf{M}'}, \widetilde{\mathbf{v}}^\top\widehat{\mathbf{M}'}\right]^\top.$$

Notice that

$$\widetilde{\mathbf{v}}^\top\widehat{\mathbf{M}'} = \frac{\left(\widehat{\boldsymbol{\Sigma}}^{-1}(\widehat{\mathbf{m}}'_k - \bar{\mathbf{x}}')\right)^\top\widehat{\mathbf{M}'} - \left(\widehat{\mathbf{B}}^\top(\widehat{\mathbf{m}}'_k - \bar{\mathbf{x}}')\right)^\top\left(\widehat{\mathbf{B}}^\top\widehat{\mathbf{M}'}\right)}{\sqrt{(\widehat{\mathbf{m}}'_k - \bar{\mathbf{x}}')^\top\left(\widehat{\boldsymbol{\Sigma}}^{-1}(\widehat{\mathbf{m}}'_k - \bar{\mathbf{x}}')\right) - \left(\widehat{\mathbf{B}}^\top(\widehat{\mathbf{m}}'_k - \bar{\mathbf{x}}')\right)^\top\left(\widehat{\mathbf{B}}^\top(\widehat{\mathbf{m}}'_k - \bar{\mathbf{x}}')\right)}}$$

and $\widehat{\mathbf{B}}^\top(\widehat{\mathbf{m}}'_k - \bar{\mathbf{x}}')$ is just the $k$th column of $\widehat{\mathbf{B}}^\top\widehat{\mathbf{M}'}$. The most time consuming computations for the matrix $[\widehat{\mathbf{B}}, \widetilde{\mathbf{v}}]^\top\widehat{\boldsymbol{\Gamma}'}[\widehat{\mathbf{B}}, \widetilde{\mathbf{v}}]$ are $\widehat{\mathbf{B}}^\top\widehat{\mathbf{M}'}$ and $\widehat{\boldsymbol{\Sigma}}^{-1}(\widehat{\mathbf{m}}'_k - \bar{\mathbf{x}}')$ which have a complexity of $O(pKH + p^2)$.

The complexity of the eigen-decomposition in (12) is $O((K+1)^3)$ and it requires $O(p(K+1)^2)$ to update $\widehat{\mathbf{B}}$ to $\widehat{\mathbf{B}}'$. So the computational complexity for each ISIR update is $O(p^2+pKH+pK^2+K^3)$. For a high dimensional problem, this is much smaller than the complexity $O(p^3 + p^2 n)$ of SIR.

## 5 Refinement by overlapping

In Zhang et al. (2018) an overlapping technique was introduced to SIR algorithm and shown effectively improving the accuracy of EDR space estimation. It is motivated by placing each observation in two or more adjacent slices to reduce the deviations of the sample slice means $\widehat{\mathbf{m}}_h$ from the EDR subspace. This is equivalent to using each observation two or more times. In this section, we adopt the overlapping technique to ISIR algorithm above to develop an incremental overlapping sliced inverse regression (IOSIR) algorithm and wish it refines ISIR.

To apply the overlapping idea, we use each observation twice. So when we have $n$ observations, we duplicate them and assume we have $N = 2n$ observations. When a new observation $(\mathbf{x}_0, y_0)$ is received, we duplicate it and assume we receive two identical observations. Based on the $y_0$ value we place the first copy into the slice $s_k$ if $\bar{y}_k$ is closest to $y_0$ and run ISIR update as described in Section 4. Note that if $\bar{y}_1 < y_0 < \bar{y}_H$, then $y_0$ must fall into the interval $[\bar{y}_{k'}, \bar{y}_k]$ with $k' = k - 1$ or it falls into $[\bar{y}_k, \bar{y}_{k'}]$ with $k' = k + 1$. So we place the second copy of the new observation into the slice $s_{k'}$, an adjacent slice to $s_k$, and run ISIR algorithm again. If $y_0 \leq \bar{y}_1$ or $y_0 > \bar{y}_H$, the second copy will be still placed into $s_k$ to guarantee all observations are weighted equally.

As OSIR has superior performance over SIR, we expect IOSIR will perform better than ISIR by a price of double calculation time.

We remark that SIR and ISIR can be used for both regression problems and classification problems. But since the concept of "adjacent slice" cannot be defined for categorical values (as is the case in classification problems), IOSIR can only be used for regression problems where the response variable is numeric.

## 6 Simulations

In this section, we will verify the effectiveness of ISIR and IOSIR with simulations on artificial and real-world data. Comparisons will be made between them and SIR.

### 6.1 Artificial data

In the simulations with artificial data, since we know the true model, we measure the performance by the accuracy of the estimated EDR space. We adopt the trace correlation $r(K) = \mathrm{trace}(\mathbf{P_B}\mathbf{P}_{\widehat{\mathbf{B}}})/K$ used in Ferré (1998) as the criterion, where $\mathbf{P_B}$ and $\mathbf{P}_{\widehat{\mathbf{B}}}$ are the projection operators onto the true EDR space $\mathbf{B}$ and the estimated EDR space $\widehat{\mathbf{B}}$, respectively.

We consider the following model from Li (1991)

$$y = x_1(x_1 + x_2 + 1) + \epsilon, \tag{13}$$

where $\mathbf{x} = [x_1, x_2, \ldots, x_p]^\top$ follows a multivariate normal distribution, $\epsilon$ follows standard normal distribution and is independent of $\mathbf{x}$. It has $K = 2$ effective dimensions with $\boldsymbol{\beta}_1 = (1, 0, 0, \ldots, 0)^\top$ and $\boldsymbol{\beta}_2 = (0, 1, 0 \ldots, 0)^\top$. We conduct the simulation in $p = 10$ dimensional space and select the number of slices as $H = 10$. We give the algorithm a warm start with the initial guess of the EDR space obtained by applying SIR algorithm to a small data set of $40$ observations. Then a total of $400$ new observations will be fed to update the EDR space one by one. SIR, ISIR, and IOSIR are applied when each observation was fed in and we calculate their trace correlation and cumulative computation time. We repeat this process $100$ times. The mean trace correlations of all three methods are reported in Figure 1(a) and the mean cumulative time is in Figure 1(b). We see that ISIR performs quite similar to SIR. IOSIR slightly outperforms ISIR and SIR. ISIR is much faster than SIR. IOSIR gains higher accuracy by sacrificing on computation time. This verifies the convergence and efficiency of ISIR and IOSIR.

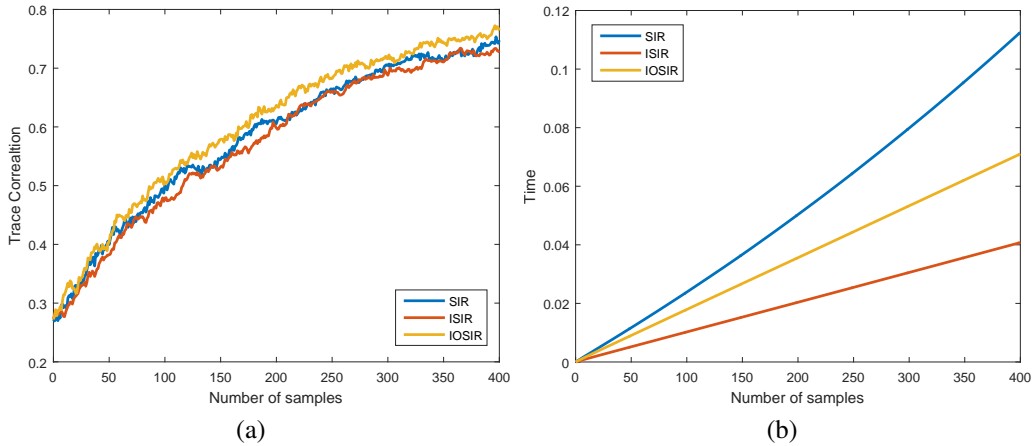

Figure 1: Simulation result for the model (13). (a) Trace correlation and (b) cumulative calculation time by SIR, ISIR, and IOSIR.

## 6.2 REAL DATA APPLICATION

We validate the reliability of ISIR on two real data sets: the Concrete Compressive Strength and Cpusmall (available on `https://www.csie.ntu.edu.tw/~cjlin/libsvmtools/datasets/regression.html`). There have been many proposed algorithms to increase the prediction accuracy on these data sets (Yeh, 1998; 1999; 2007; Öztaş et al., 2006; Gu et al., 2015). We do not intend to outperform those methods. Our goal is to compare the performance of supervised dimension reduction algorithms and verify the effectiveness of our incremental methods.

The Concrete Compressive Strength data has $p = 8$ predictors and 1030 samples. We use $H = 10$ and $K = 3$ to run SIR, ISIR, and IOSIR. We select 50 observations to warm start ISIR and IOSIR algorithms, then 700 observations are fed sequentially. The left 280 observations are used as test data. After each new observation is received we estimate the EDR space, project the available training set to the estimated EDR space, build a regression model using the k-nearest neighbor method, and compute the MSE on the test data set. This process is repeated 100 times and the average MSE was reported in Figure 2(a). For the Cpusmall data, which has $p = 12$ predictors and 8192 samples, we do the experiment with $H = 10$, $K = 3$, 50 observations to warm start ISIR and IOSIR, 2000 observations for sequential training, and 6142 observations for testing. The average MSE was plotted in Figure 2(b). The results indicate both ISIR and IOSIR are as effective as SIR.

## 7 CONCLUSIONS AND DISCUSSIONS

We proposed two online learning approaches for supervised dimension reduction, namely, ISIR and IOSIR. They are motivated by standardizing the data and reformulate the SIR algorithm to a PCA problem. However, data standardization is only used to motivate the algorithm while not explicitly calculated in the algorithms. We proposed to use Sherman Morrison formula to online update $\mathbf{\Sigma}^{-1}$ and some approximated calculations to circumvent explicit data standardization. This novel idea played a key role in our algorithm design. Both algorithms are shown effective and efficient. While IOSIR does not apply to classification problems, it is usually superior over ISIR in regression problems.

We remark that the purpose of ISIR and IOSIR is to keep the dimension reduction accuracy in the situation that a batch learning is not suitable. This is especially the case for streaming data where information update and system involving is necessary whenever new data becomes available. When the whole data set is given and one only needs the EDR space from batch learning, ISIR or IOSIR is not necessarily more efficient than SIR because their complexity to run over the whole sample path is $O(p^2 N)$, comparable to the complexity $O(p^3 + p^2 N)$ of SIR.

There are two open problems worth further investigation. First, the need to store and use $\mathbf{\Sigma}^{-1}$ during the updating process is the main bottleneck for ISIR and IOSIR when the dimensionality of

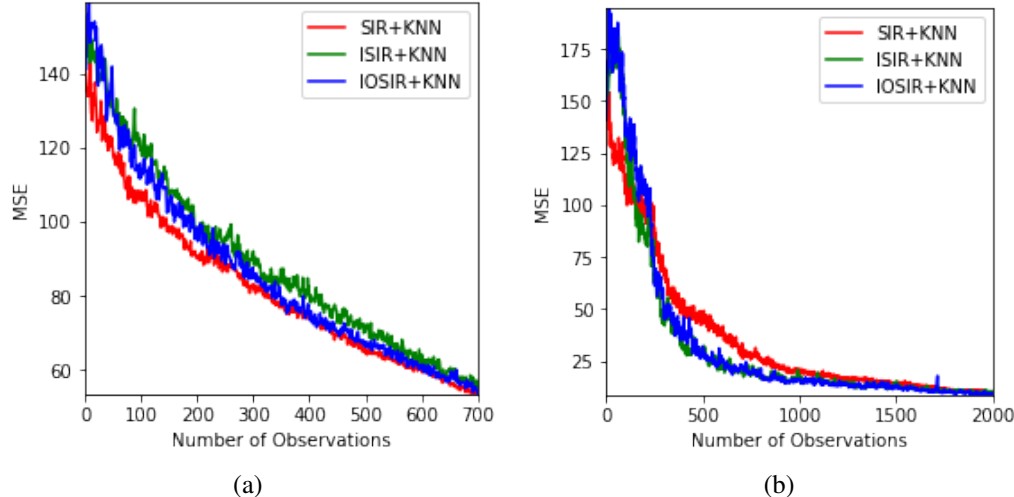

Figure 2: Mean square errors (MSE) for two real data applications: (a) for Concrete Compressive Strength data and (b) for Cpusmall data.

the data is ultrahigh. Second, for SIR and other batch dimension reduction methods, many methods have been proposed to determine the intrinsic dimension $K$; see e.g. Li (1991); Schott (1994); Bura & Cook (2001); Bai & He (2004); Barrios & Velilla (2007); Nkiet (2008). They depend on all $p$ eigenvalues of the generalized eigen-decomposition problem and are impractical for incremental learning. We do not have obvious solutions to these problems at this moment and would like to leave them for future research.

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
