# OpenReview forum: "Online Learning for Supervised Dimension Reduction"
_ICLR.cc/2019/Conference_

### Official Review · AnonReviewer2 · 2018-10-30
**Algorithm directly motivated by online PCA**

**Rating:** 6
**Confidence:** 5

**Review:**

Sliced Inverse Regression is a well-known technique for finding EDR space in supervised dimension reduction problems, under condition that input X is normally disributed. When the number of dimensions is large and an access to observations is online, finding eigenvalues of covariance matrices (online) could become computationally costly. So, the paper focuses the problem of updating few principal components of covariance matrices as new examples come.

The key idea is that in classical SIR two problems are solved sequentially (estimation of Sigma = cov(X) and finding PCA of vectors Sigma^{-0.5}X_s where s corresponds to average X of s-th slice). The second part can be reduced to an online version of PCA (Peter M Hall et al.).

The paper seems consistent and experiments convince that the proposed method works. The lack of theoretical analysis is a disadvantage.

Also, some typos:
1. Γb = λΣbβ (p.4 par.3) --- beta omitted
2. We can regard z0 = Σ^{-1/2}m'_k (p.4) --- bar over m omitted
3. Formula 10 looks weird (Sigma instead of Sigma prime)

---

### Official Review · AnonReviewer1 · 2018-11-02
**the algorithm is not efficient for large p**

**Rating:** 5
**Confidence:** 4

**Review:**

This paper proposes an online learning algorithm for supervised dimension reduction, called incremental sliced inverse regression (ISIR). The key idea is converting the SIR problem into PCA problem by using the inverse of covariance matrix. After the transformation, we can use incremental PCA to compute the top eigenvector and obtain the approximate solution of SIR in streaming way. The authors also extend ISIR to overlapping case.

The motivation of this paper is reasonable, but I have some concerns as follow.

1. The computation of ISIR is dependent on maintaining the matrix \hat \Sigma’ (or its inverse), which requires O (p^2) time and space. In my opinion, this complexity is too expensive for high dimensional datasets which makes the main result of this paper is not strong. Maybe we can use low-rank approximation and its variants to improve the efficiency.

2. For large dataset, the covariance matrix may be ill-conditioned with more and more data arriving even we use warm start strategy at first. It is more reasonable to introduce a ridge term to make the algorithm more stable.

3. The experiments only evaluate on some small datasets. It is not enough to show the advantage of the proposed algorithms. There are also many other strategy can be used into this problem such as random sampling, random projections and frequent directions, but this paper does not provide sufficient discussion.

---

### Official Review · AnonReviewer4 · 2018-11-09
**Rebadging of Incremental Generalized Eigenvalue Decomposition.**

**Rating:** 2
**Confidence:** 5

**Review:**

This paper studies sufficient dimension reduction problem, and proposes an incremental sliced inverse regression algorithm. Numerical experiments are provided to demonstrate the effectiveness of the proposed algorithms.

The sliced inverse regression here is nothing but generalized eigenvalue decomposition:

Ax=lambda Bx.

Note that Multiclass Fisher Linear Discriminant Analysis, Canonical Correlation Analysis, Nonlinear Manifold Embedding and many subspace learning methods can also be formulated as generalized eigenvalue decomposition. All these methods need to compute covariance-like matrices in the additive form, which makes incremental update very convenient.

The incremental generalized eigenvalue decomposition has been extensively studied for over decades, especially between 1995 and 2005 in the face recognition community. I am just listing a few here:

Ye et al., IDR/QR: An Incremental Dimension Reduction Algorithm via QR Decomposition, 2005

Law and Jain, Incremental Nonlinear Dimensionality Reduction by Manifold Learning, 2006

Yan et al. Towards incremental and large scale face recognition, 2011

Ghassabeh et al. A New Incremental Face Recognition System, 2007

Song et al. A Novel Supervised Dimensionality Reduction Algorithm for Online Image Recognition, 2006.

Wang et al. Incremental two-dimensional linear discriminant analysis with applications to face recognition, 2010.

Salman et al. Efficient update of the covariance matrix inverse in iterated linear discriminant analysis, 2010

Park and Park, A comparison of generalized linear discriminant analysis algorithms, 2008

Wang, INCREMENTAL AND REGULARIZED LINEAR DISCRIMINANT ANALYSIS, 2012

These algorithms become less popular/known now, because (1) they are not scalable and efficient for large p, and (2) these classical dimensionality reduction methods perform poorly in many tasks, compared with the state of art results.

This paper only cites a few papers on incremental LDA,  but does not even mention that both LDA and SIR are essentially solving similar optimization problems. Moreover, it does not compare the results with any of the above references, either.

This paper even claims applying the Sherman–Morrison formula as the contribution. However, such an  update has been used in Salman et al. 2010, Park and Park 2008, Wang 2012.

In summary, this paper is far below the bar of ICLR.

Minor: There are numerous typos in this paper. The authors even misspell "Morrison" in the Sherman–Morrison formula as "Morison".

---

### Meta-Review · Area_Chair1 · 2018-12-17
**Contribution unclear with respect to substantial relevant literature**

**Confidence:** 5
**Recommendation:** Reject

**Metareview:**

The paper investigates an incremental form of Sliced Inverse Regression (SIR) for supervised dimensionality reduction.  Unfortunately, the experimental evaluation is insufficient as a serious evaluation of the proposed techniques.  More importantly, the paper does not appear to contribute a significant advance over the extensive literature on fast generalized eigenvalue decompositions in machine learning.  No responses were offered to counter such an opinion.